

# Coastal fish assemblages and predation pressure in northern-central Chilean *Lessonia trabeculata* kelp forests and barren grounds

Nicolás Riquelme-Pérez[1], Catalina A. Musrri[1], Wolfgang B. Stotz[1], Osvaldo Cerda[1], Oscar Pino-Olivares[1] and Martin Thiel[1,2,3]

[1] Facultad de Ciencias del Mar, Universidad Católica del Norte, Coquimbo, Chile
[2] Millennium Nucleus Ecology and Sustainable Management of Oceanic Island (ESMOI), Coquimbo, Chile
[3] Centro de Estudios Avanzados en Zonas Áridas (CEAZA), Coquimbo, Chile

## ABSTRACT

Kelp forests are declining in many parts of the globe, which can lead to the spreading of barren grounds. Increased abundances of grazers, mainly due to reduction of their predators, are among the causes of this development. Here, we compared the species richness (SR), frequency of occurrence (FO), and maximum abundance (MaxN) of predatory fish and their predation pressure between kelp forest and barren ground habitats of northern-central Chile. Sampling was done using baited underwater cameras with vertical and horizontal orientation. Two prey organisms were used as tethered baits, the black sea urchin *Tetrapygus niger* and the porcelanid crab *Petrolisthes laevigatus*. SR did not show major differences between habitats, while FO and MaxN were higher on barren grounds in vertical videos, with no major differences between habitats in horizontal videos. Predation pressure did not differ between habitats, but after 24 h consumption of porcelanid crabs was significantly higher than that of sea urchins. *Scartichthys viridis/gigas* was the main predator, accounting for 82% of the observed predation events on *Petrolisthes laevigatus*. Most of these attacks occurred on barren grounds. *Scartichthys viridis/gigas* was the only fish observed attacking (but not consuming) tethered sea urchins. High abundances of opportunistic predators (*Scartichthys viridis/gigas*) are probably related to low abundances of large predatory fishes. These results suggest that intense fishing activity on large predators, and their resulting low abundances, could result in low predation pressure on sea urchins, thereby contributing to the increase of *T. niger* abundances in subtidal rocky habitats.

# INTRODUCTION

Kelp forests are communities formed by large brown seaweeds, which can reach high densities and several meters in height (*Wernberg et al., 2019*). These ecosystems are vital to a wide diversity of invertebrate and fish species (*Steneck et al., 2002*; *Almanza et al., 2012*;

Corresponding author
Martin Thiel, thiel@ucn.cl

*Miller et al., 2018*), providing them with food, shelter, and habitat for larval settlement (*Graham, Vásquez & Buschmann, 2007*; *Teagle et al., 2017*). However, global trends indicate a decline in kelp forests (*Krumhansl et al., 2016*), as a consequence of poor habitat quality, pollution or sedimentation (*Foster & Schiel, 2010*), global warming (*Wernberg et al., 2010*), heatwaves (*Wernberg et al., 2013*; *Oliver et al., 2018*), or El Niño events (*Tegner & Dayton, 1987*; *Vega, Vásquez & Buschmann, 2005*), which affects the reproduction, settlement and physiology of kelp (*Simonson, Scheibling & Metaxas, 2015*; *Hargrave et al., 2017*). Kelp abundances also can be affected through direct extraction by humans or by increased abundances of grazers (*Tegner & Dayton, 2000*; *Steneck et al., 2002*). In the absence of these kelp forests, there are often extensive patches of rock devoid of vegetation called 'barren grounds', which have low biodiversity and high abundances of grazers, such as sea urchins and snails (*Norderhaug & Christie, 2009*; *Wernberg et al., 2019*).

Grazers, especially sea urchins, are one of the most important factors controlling the distribution of kelp forests and can consume young kelp recruits, thereby suppressing the renewal and recovery of kelp populations (*Norderhaug & Christie, 2009*; *Konar, Edwards & Estes, 2014*; *Perreault, Borgeaud & Gaymer, 2014*; *Wernberg et al., 2019*). The increase of grazer populations can be due to natural and/or anthropogenic causes, which might generate favourable conditions for the survival and growth of grazers (*Guidetti et al., 2003*; *Leleu et al., 2012*; *Konar, Edwards & Estes, 2014*). The best-documented process that affects grazer abundances is the alteration of trophic webs, with a reduced 'top-down' pressure due to decreasing predator populations, which then leads to rising grazer populations (*Halpern, Cottenie & Broitman, 2006*; *Leleu et al., 2012*; *Hamilton & Caselle, 2015*; *Medrano et al., 2019*). Furthermore, there are several examples of top-down effects in kelp forests, especially in marine reserves or marine protected areas, where predators are protected. In these cases, high predator abundances control grazers leading to increased algal biomass (*Guidetti, 2006*; *Clemente et al., 2010*; *Leleu et al., 2012*; *Selden et al., 2017*). Outside of protected areas, many of these predators are important marine resources that have been overharvested by humans (*Jackson et al., 2001*), thus indirectly leading to a higher grazer pressure.

The shallow rocky subtidal zone (0–20 m depth) of the Chilean coast is dominated by kelp forests (*Lessonia trabeculata* and *Macrocystis pyrifera*) and barren grounds, the latter characterised by high abundances of sea urchins *Tetrapygus niger* and *Loxechinus albus* and snails from the genus *Tegula* (*Dayton, 1985*; *Vásquez & Buschmann, 1997*; *Stotz et al., 2016*; *Pérez-Matus et al., 2017a*). In addition to direct human exploitation of seaweeds, these herbivorous species are also considered one of the main causes for the decrease of Chilean kelp forests and their associated fauna (*Dayton, 1985*; *Vásquez & Santelices, 1990*; *Vásquez & Buschmann, 1997*; *Henríquez et al., 2011*; *Perreault, Borgeaud & Gaymer, 2014*). This can be due to a decrease in predation pressure on sea urchins. Some fish species, such as *Pinguipes chilensis*, *Semicossyphus darwini*, *Cheilodactylus variegatus*, *Oplegnathus insignis,* and *Graus nigra*, are important predators in the shallow rocky communities of northern-central Chile (*Muñoz & Ojeda, 1997*; *Vargas, Soto & Guzmán, 1999*; *Pérez-Matus et al., 2012*, *2017b*) where they also prey on these grazer species. However, most of these fishes have been subject to an intense fishing activity with important population declines in

northern-central Chile (*Godoy et al., 2010*), possibly resulting in the high number of sea urchins (*Urriago, Himmelman & Gaymer, 2012*).

Along the Chilean coast, the effects of sea urchins on kelp have been widely studied (*Perreault, Borgeaud & Gaymer, 2014*), but the role of different coastal fish predators in controlling sea urchin populations and the effects of reduced abundances of these fishes in this system are less known. In order to fill this gap, we compared the effect of coastal fishes in kelp forests and on barren grounds in northern-central Chile by surveying fish assemblages and estimating predation on tethered live prey organisms in these contrasting habitats. Kelp forests are structurally complex systems that tend to have greater richness and abundance of predatory species compared to less complex systems (*Hauser, Attrill & Cotton, 2006*; *Miller et al., 2018*; *Villegas et al., 2018*) such as barren grounds (*Sala & Zabala, 1996*; *Norderhaug & Christie, 2009*). Therefore, we expected to find higher species richness (SR) and abundances of predatory fish in kelp forests than on barren grounds, and hence higher predation pressure on our baits at sites with a greater presence of predators.

## MATERIALS AND METHODS

### Study location

The study was conducted at four locations in northern-central Chile (28°S–30°S) between September 2016 and October 2017 (Table S1; Fig. 1). Study locations were Caleta Angosta (CA), Punta Choros (PC), Chungungo (CH), and Guayacancito (GU), which are located within the biogeographic province north of latitude 30°S, considered as an important biogeographical break (*Camus, 2001*). These locations were chosen mainly because of the presence of one or both communities of interest (kelp forests and barren grounds) and accessibility to the sites. Kelp forests were mainly composed of *Lessonia trabeculata* with a minimum extension of 100 m along the shoreline and had at least a 1-year absence of algal harvesting. They were extensive forests with densities that ranged from 0.5 ind. m$^2$ (PC and CH) to 2 ind. m$^2$ (CA). Barren grounds were composed of open rock fields, including bedrock and boulders.

Sampling was done at an average depth of 10 m on all sites. All study locations were exposed to the open sea. Data were collected in different seasons but considering that coastal fishes present in the study area are mainly non-migratory species (*Angel & Ojeda, 2001*; *Pérez-Matus et al., 2007*) and there are only minor seasonal variations among predatory fish assemblages (*Pérez-Matus et al., 2007*, *2012*), findings are comparable over time.

Two locations had both types of habitat (PC and CH), and kelp forests and barren grounds were separated by approximately 500 m at PC and by ~2,000 m at CH (Fig. 1). PC and CH share general characteristics such as (a) distance to large urban centres (moderate potential for anthropogenic disturbances), (b) complex morphology and architecture of the coast and substratum, and (c) administrative regimes under the framework of Management Areas for Benthic Resources (AMERBs for Spanish abbreviation). There were also two distant locations with contrasting habitats, one with a large kelp forest (CA) located at the northern edge of our study area, and the other with an extensive barren ground (GU) located to the south. CA also is located within an AMERB, far from urban

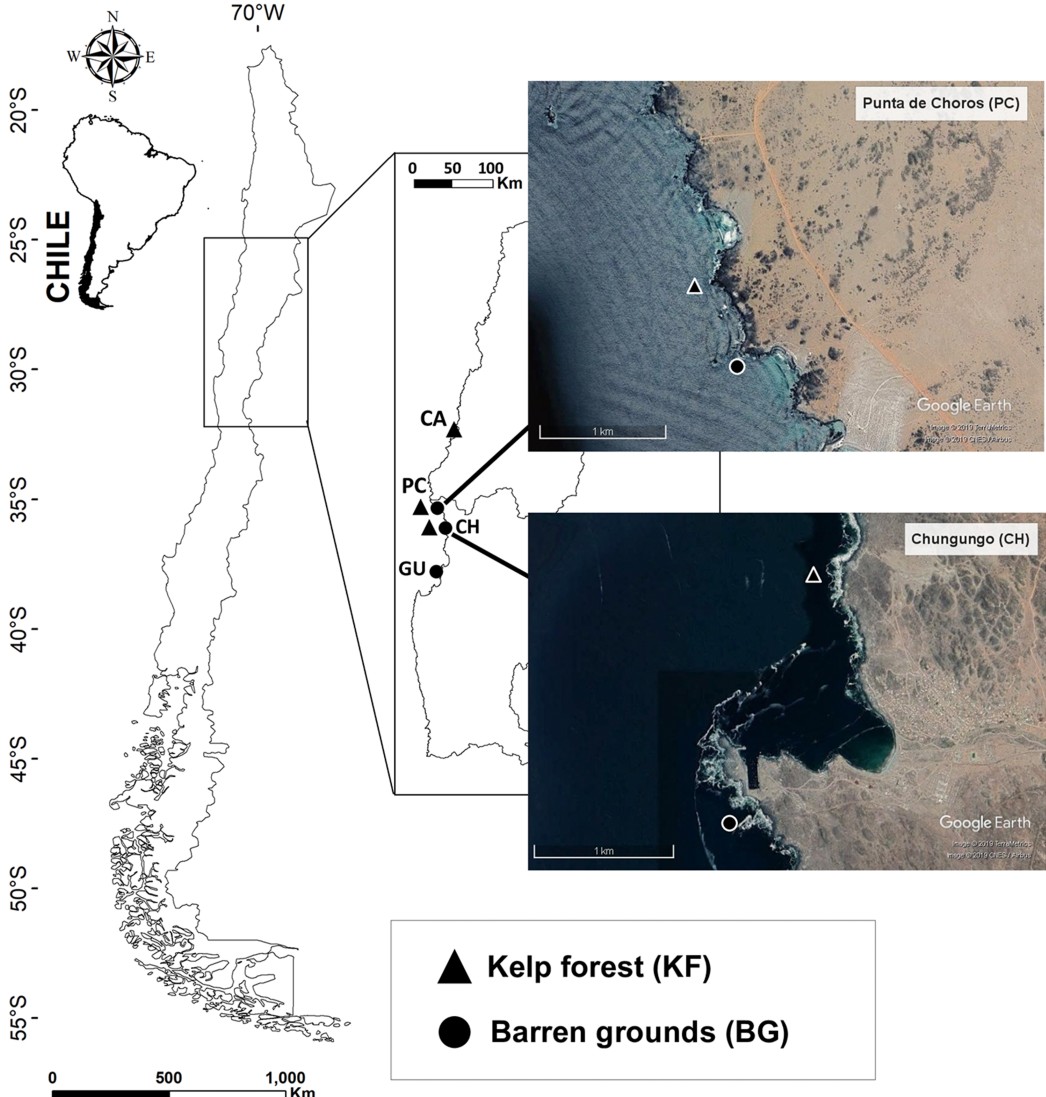

**Figure 1 Map of the four sampling locations along the Chilean coast and their corresponding habitats.** Black triangles indicate kelp forest sites at Caleta Angosta (CA), Punta de Choros (PC), and Chungungo (CH) and black dots indicate barren ground sites at PC, CH, and Guayacancito (GU). All sites are within management areas (AMERBs), except for GU, which is an open access area. Images were extracted from Google Earth Pro program. Image © 2018 Google, CNES/airbus 2019, Terrametrics 2019.

centres, while GU is an open access area close to Coquimbo, a major urban centre. These locations with non-paired habitat types were included because they belong to coastal stretches of several kilometre with extensive coverage of the respective habitat type (either kelp forest or barren ground). Along the coast of northern-central Chile, it is difficult to find an ideal kelp forest paired with an adjacent barren ground since some areas are heavily exploited for kelps (within and outside of AMERBs), leading to extensive barren grounds. On the other hand, in other areas kelp forests have remained untouched for many years (mostly due to logistic reasons impeding exploitation) and only small patches without kelps are found.

To facilitate the following descriptions, we term each habitat at each location a 'study site', and so we have three sites for kelp forest community (KF-CA, KF-PC, KF-CH) and three sites for barren ground community (BG-PC, BG-CH, BG-GU).

## Experimental design

Fish surveys were conducted with baited remote underwater videos (BRUVs), which allowed us to determine fish abundances, predation pressure, and identity of predators. BRUVs consisted of two different types of structures with GoPro cameras in vertical and horizontal orientation, respectively. The first structure (vertical orientation) was a metal frame with the camera fastened in the upper part, which allowed images to be taken from 80 cm above the bottom (vertical view) of an approximate area of 0.6 m$^2$ (Figs. 2A and 2C). The second structure (horizontal orientation) was a plate with a metal arm to which the camera was fastened, which took panoramic images (horizontal view) where the bottom area was measured as a trapezium (Figs. 2B and 2D). Each structure had on its base an eternit fibre cement plate of 40 × 40 cm side length, to which prey items were tethered. Each plate contained six tethered juveniles of the black sea urchin *T. niger* or six juveniles of the porcelanid crab *Petrolisthes laevigatus*, respectively (see details below). Both types of structures were held on the bottom by concrete pieces of one kg or mesh filled with stones and a small and numbered plastic bottle was attached with a thin rope to each structure, allowing us to rapidly locate them from the boat.

Two different types of structures were used due to their respective advantages and/or shortcomings: vertical structures, with a small area, allowed more precise images (because the camera is closer to the plate), which is very useful to identify fish species and to observe predation events on tethered baits when visibility is low (see also *Schettini & Corchs, 2010*), but the vertical videos do not allow to observe the surrounding community; on the other hand, horizontal structures produce a greater, but more imprecise observational area, allowing to see what happens in the surrounding community during the assay period, although high turbidity might occasionally limit visibility. Indeed, most studies that use cameras to describe subtidal fish assemblages have the cameras oriented in the horizontal plane (*Malcolm et al., 2007*; *Duffy et al., 2015*).

At each study site (KF-CA, KF-PC, KF-CH, BG-PC, BG-CH, BG-GU), five vertical and five horizontal structures were randomly distributed for each prey (not all of them with camera, because of camera availability). Deployment of the assays was repeated over two different days, because only five structures were available of each type (vertical and horizontal; see Table S2). So, after the two deployment days there were five replicates with *Petrolisthes laevigatus* on horizontal structures and five replicates with *Petrolisthes laevigatus* on vertical structures, and the same for *T. niger*. This was done during consecutive days for CA and PC, but not for CH and GU, due to logistic reasons (e.g. poor weather conditions delaying the continuation of field work for a few weeks). However, since fish assemblages do not vary much over the year (*Pérez-Matus et al., 2007*, *2012*), this should not affect the results. Assays were deployed between 9:00 am and 11:00 am, at ~10 m depth (at sites with abundant kelp coverage in the case of kelp forest habitats, see Fig. S1), and separated from each other by similar distances (~10 m), following a line

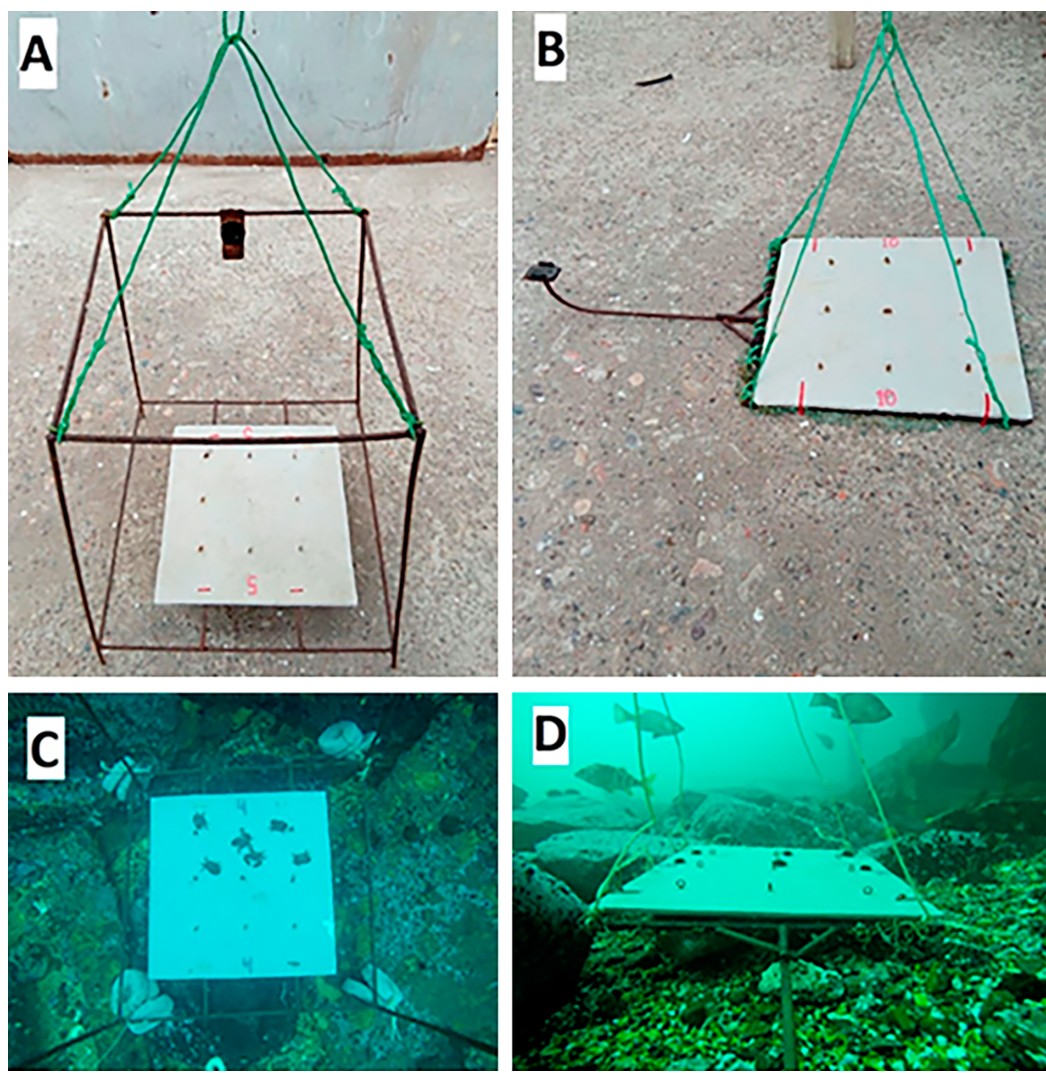

**Figure 2 Structures used to record communities, to carry out predation assays and to document predation events.** (A) Metal frame structure to generate vertical view, (B) metal arm structure to generate horizontal view, (C) vertical view perpendicular to the bottom, and (D) horizontal view parallel to the bottom. Square structures on the base of the structures were eternity fibre cement plates of 40 × 40 cm. Photo Credits: Nicolás Riquelme–Pérez.

parallel to the coast. The distance of 10 m between structures is considered sufficient to prevent the directional movement of fishes from one structure to the next. The fish species observed in our study move randomly and the most frequent fishes are small and territorial species, and thus movements between structures are unlikely and each structure can be considered an independent replicate. Furthermore, visibility in the system is limited due to kelp fronds, complex habitat structures with large boulders or rocky outcrops, and high turbidity, thus limiting visibility from one structure to the next. Field work was conducted under the resolution 2,649 from 'Subsecretaría de Pesca y Acuicultura' published in the 'Diario Oficial de la Republica de Chile', num. 41,553 on 7 September 2016.

## Fish species richness, frequency of occurrence, and maximum abundance

We used video recordings to compare SR, frequency of occurrence (FO), and maximum abundance (MaxN) of fish between habitats, and to determine the total number of species (total SR) per site. In each recording, we registered fish species and their abundances every 30 s (video frame). When it was not possible to adequately scan a video frame (e. g. kelp blades in front of the camera), the closest visible frame was analysed. For horizontal videos, only fish observed within <2 m distance from the camera (distance estimates based on reference points of the structures) were recorded, thereby generating a standardised survey area for all videos; there were no videos with a visibility of <2 m. We obtained a total of 81 videos for both habitat and structure types (for details, see Table S2).

Species richness was the number of species per video for each study site, FO was calculated as the proportion of frames in which the respective species was observed over the total number of frames analysed in each video (80 frames; providing information about residence or staying time), and MaxN was the maximum number of individuals of the respective species in a video (recorded in one of the 80 examined frames, that is, there is only one value of MaxN per species per video), which provides information about the relative abundance of the given species at the study site (*Cappo et al., 2003*). All fishes were identified to species based on their shape, colour, and swimming habits, with the exception of *Scartichthys*, of which two species occur in the study area, *Scartichthys viridis* and *Scartichthys gigas*. These species are common in northern Chile and display variable colour patterns (*Méndez-Abarca & Mundaca, 2016*), making it sometimes difficult to distinguish among them, especially in videos with limited visibility. As both species are of similar size and ecological habits, we treated them as *Scartichthys viridis/gigas* (see also *Villegas et al., 2018*).

To standardise the recordings, only the initial 40 min of each video were used (see also *Willis & Babcock, 2000*), obtaining 80 frames for counting data. This standardization was done for the analysis of SR, FO, and MaxN, all with one single data point per video for analyses. Species accumulation curves for the two types of habitat confirmed that mean number of species recorded with the videos approached an asymptote before 40 min (Fig. S2), a time frame also confirmed for a similar study system by *Harasti et al. (2015)*. Only total SR and predation events were assessed using the entire duration of the videos.

## Predation assays and identity of predators

We measured predation pressure using tethered prey organisms (*T. niger* and *Petrolisthes laevigatus*) as bait. Tethering has been used in different types of communities and with various types of prey, including snails (*Reynolds et al., 2018*), crustaceans (*Heck & Wilson, 1987*; *Gutow et al., 2012*; *Ory et al., 2012*), sea urchins (*Urriago, Himmelman & Gaymer, 2011*, *2012*), and fishes (*Pérez-Matus et al., 2016*). Since refuge availability, prey behaviour, and other variables can affect the outcome of tethering assays, these should only be used to assess relative predation or predation pressure within the same ecosystem; following those basic recommendations (as done in our study) tethering assays continue to be a useful tool

for comparing the predation rates among different habitats or environments (*Aronson & Heck, 1995*).

As small sea urchins tend to be susceptible to predation (*McClanahan, 1988*), juvenile sea urchins *T. niger* (15–25 mm test diameter) were tethered with monofilament fishing line (0.15 mm diameter) according to the methodology used by *McClanahan & Muthiga (1989)* and *McClanahan & Shafir (1989)*. The porcelanid crab *Petrolisthes laevigatus* was included in our study for comparative purposes, because along the Chilean coast decapod crustaceans are more frequent prey than sea urchins in the diet of common fish predators from subtidal hard bottoms (*Pérez-Matus et al., 2012*). Thus, this crab was used to validate background predation in the study area. Individuals of *Petrolisthes laevigatus* (15–20 mm of cephalothorax) were tethered by gluing a piece of monofilament to the carapace with cyanoacrylate glue. Preliminary experiments of 24 h duration were run in laboratory tanks with tethered individuals, and neither mortality (because of the tether) nor escapes were observed.

Prey organisms were tethered on eternit plates to provide a uniform surface to prey and predators across different types of habitats; this way different refuge structures and prey hiding behaviour in the two habitat types would not interfere with the estimates of predation pressure. After 24 h, missing prey items were counted for each structure and divided by the total number of baits ($n = 6$ per structure), obtaining the proportion consumed. Using video recordings, we also identified predator species that consumed either sea urchins or crabs. It should be noted that data about predation events (from videos) only included the period until the cameras stopped recording (after ~40–60 min), and we may not have identified all species that might have fed on our prey organisms during the 24 h of consumption assays.

## Data analyses

We analysed fish SR, FO, and MaxN separately for vertical and horizontal structures, considering the differences in the view field of each structure. The predation data after 24 h were pooled for both structures, because the plate areas with attached prey organisms were the same, irrespective of the structure.

For analysis, data from the two sampling days were pooled (see also above). All procedures were performed through stepwise model selection on the raw data, according to methods from *Zuur et al. (2009)*. SR data were contrasted between habitats and sites using generalised linear models (GLM), pooling video data from both prey species; consumption was contrasted between habitat and prey using GLM, pooling video data from the two different structures; and FO and MaxN were contrasted between habitat and prey types using generalised linear mixed models (GLMMs), considering the values of FO or MaxN, respectively, of each species per video as duplicates. GLMMs allowed us to account for our unbalanced, over-dispersed and zero-inflated data set (*Zuur et al., 2009*). In GLMM analyses, we included habitat and prey type as fixed factors; and sites, fish species, and individual observations (replicates) were included as random factors. Zero inflation structure in FO GLMMs was attributed to habitat and prey type, whereas in MaxN GLMMs it was only attributed to habitat. Sample replicates were included as

observational-level random effects in order to account for overdispersion, as recommended by *Bolker et al. (2009)*. The best model fit was selected based on the lowest corrected Akaike Information Criterion (AICc) value. Model validation was then evaluated through graphical observations of the normality of residuals and the relationship between both observed and standardised residuals against fitted values (*Bolker et al., 2009*; *Zuur et al., 2009*).

All best-fit model validations exhibited an acceptable relationship of residuals against fitted values. Most residuals were normally distributed, while observed residual against fitted values did not show any pattern (Table S3). However, subtle heterogeneities in standardised residuals against fitted values were observed in some cases. Additional structural-zeros were not predicted for any of the final best-fit models.

Statistical analyses and graphics were done using the R environment (*R Development Core Team, 2018*), and its libraries glmmTMB (*Brooks et al., 2017*), AICcmodavg (*Mazerolle, 2019*), DHARMa (*Hartig, 2018*), ggplot2 (*Wickham, 2016*), and MASS (*Venables & Ripley, 2002*).

## RESULTS

### Fish species richness, frequency of occurrence and maximum abundance

We observed a total of 11 fish species analysing the entire video duration, 10 species in kelp forests and eight on barren grounds (Fig. 3). A species that was absent in kelp forests was *Paralabrax humeralis*, which was only observed at BG-PC, while the following species were not recorded on barren grounds: *G. nigra*, *Semicossyphus darwini*, and *Acanthistius pictus*. The latter two were only observed at KF-CA. The highest total number of species (nine species) were observed in the CA kelp forest, while only four species were seen on the GU barren ground. For PC and CH, where both habitats were present, BG and KF had similar total SR (six and five species, respectively, in both locations, Fig. 3).

Species richness (mean ± standard deviation) ranged from 1.2 ± 0.4 fish species in the CH kelp forest to 2.8 ± 1.6 in the CA kelp forest. The low values of SR were due to videos in which only one or two species were observed (see the dots in Fig. 4). SR was similar between locations and habitats, both for videos with vertical and horizontal view (Fig. 4; Table S4).

*Scartichthys viridis/gigas* was the most abundant species, followed by *Pinguipes chilensis* and *Cheilodactylus variegatus*, in both vertical and horizontal videos (Fig. 5). *Scartichthys viridis/gigas* and *Pinguipes chilensis* used to be the first species to arrive in the visual field of the cameras (in both habitats). It was very common to see these species within seconds after the structure was deployed, or even when the diver was still there. Other, less common species were observed in videos within minutes after deployment.

For vertical videos, there were no differences in FO between prey types, although barren grounds were significantly different from kelp forests (Wald $z$ = 4.625, d$f$ residual = 359, $p$ < 0.001). In addition, random effects were largely determined by fish species variance (Table S5). For horizontal videos, we also found no differences in FO between prey types and FO was higher on barren grounds (Wald $z$ = 2.311, d$f$ residual = 341, $p$ = 0.0208).

| Species | CA KF | PC KF | PC BG | CH KF | CH BG | GU BG |
|---|---|---|---|---|---|---|
| *Scartichthys viridis/gigas* | ● | ● | ● | ● | ● | ● |
| *Pinguipes chilensis* | ● | | ● | ● | ● | ● |
| *Cheilodactylus variegatus* | ● | ● | ● | ● | ● | ● |
| *Aplodactylus punctatus* | ● | ● | ● | ● | | |
| *Chromis crusma* | | ● | ● | | ● | ● |
| *Labrisomus philippii* | ● | ● | | | ● | |
| *Hemilutjanus macrophthalmos* | ● | | | | ● | |
| *Graus nigra* | ● | | | ● | | |
| *Paralabrax humeralis* | | | | ● | | |
| *Semicossyphus darwini* | *● | | | | | |
| *Acanthistius pictus* | *● | | | | | |
| **Total fish species richness:** | 9 | 5 | 6 | 5 | 6 | 4 |
| **Number of videos (Vertical/Horizontal):** | 7/5 | 9/9 | 6/8 | 5/6 | 8/6 | 6/6 |

**Figure 3 Species of fish observed in videos by location and habitat (KF = Kelp forest, BG = Barren grounds), black fish shapes show presence.** Fish species richness is the number of fish species observed at the respective location and habitat. *Species observed outside the 40 min standardised video duration for quantitative evaluation.

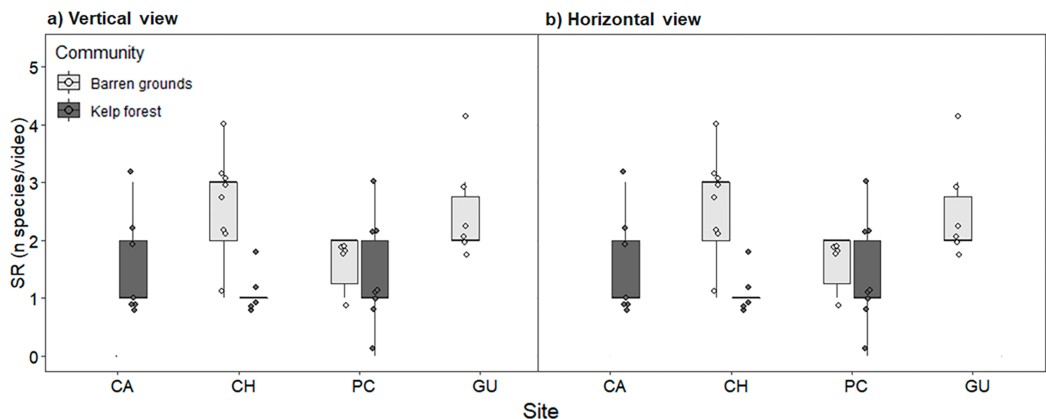

**Figure 4 Fish species richness (SR, number of species video$^{-1}$) by study site, observed in videos with (A) vertical, and (B) horizontal view.** During the first 40 min of each video, one frame was scanned every 30 s, adding up to a total of 80 frames that were considered for analyses. The box plots present first and third quartile, while the bold lines in the boxes show the medians of SR. Vertical lines are the minimum and maximum values (1.5*IQR, interquartile range); dots show jittered raw data points and dots outside the range are outliers.

Random effects were again determined by fish species variance (Table S6). For the FO of three species that generated the greatest random effect, *Scartichthys viridis/gigas* and *Pinguipes chilensis* reached higher values on barren grounds (mainly in vertical videos),

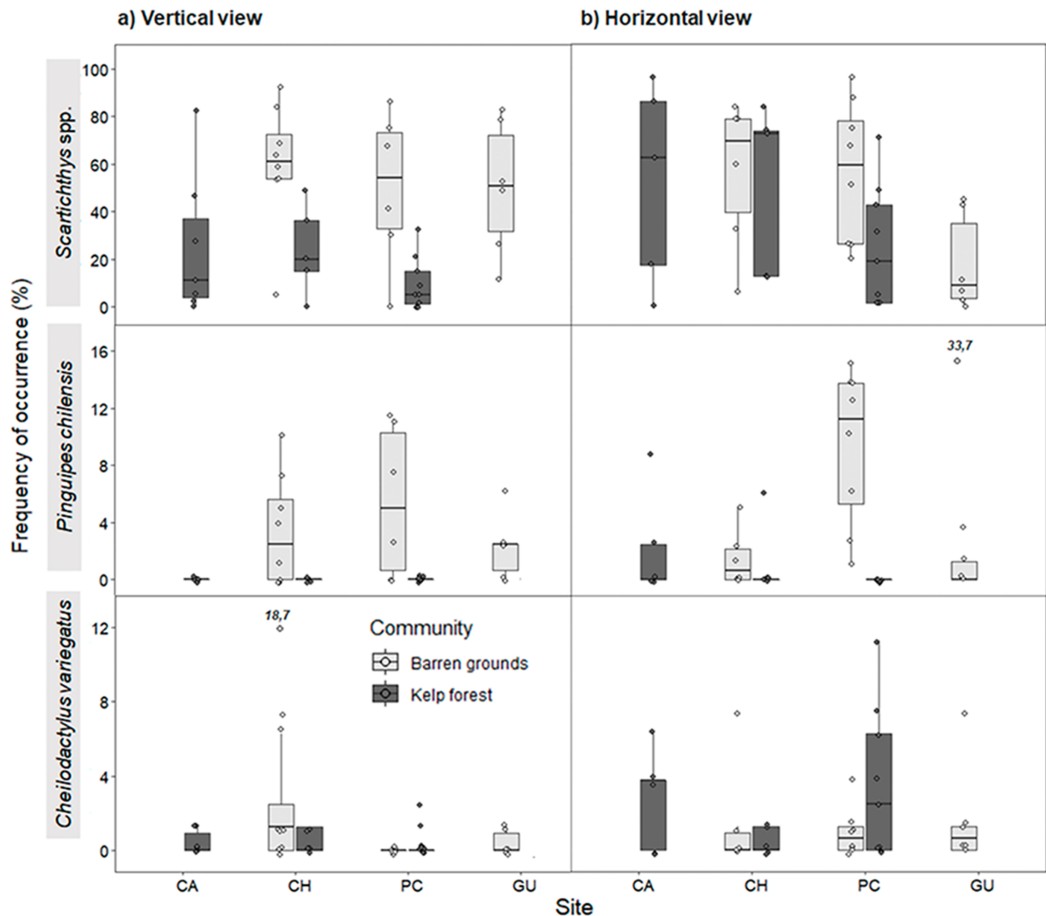

**Figure 5 Frequency of occurrence (%) (FO) of the three most common predatory fish species by study site in videos with (A) vertical, and (B) horizontal view.** FO was percentage of frames in which at least one individual of each species was observed (of a total of 80 frames that were scanned every 30 s during the first 40 min of the video). The box plots present first and third quartile, while the bold lines in the boxes show the medians of FO; vertical lines are the minimum and maximum values (1.5*IQR or the interquartile range); dots show jittered raw data points and dots outside the range are outliers; dots with a number above them (which correspond to a FO value), represent outliers outside the y-axis scale.

whereas *Cheilodactylus variegatus* mostly had higher values of FO in kelp forests (Fig. 5). Other predatory species, although present, occurred very infrequently, seen only in one video or in few frames.

Maximum abundance for the same three most abundant predatory fish species followed a similar pattern as for FO, both in vertical and horizontal videos. For vertical videos, prey type did not exhibit a significant effect, although barren grounds were significantly different from kelp forests (Wald $z = 4.793$, df residual = 360, $p < 0.001$). Random effects were again largely determined by fish species variance (Table S7). For horizontal videos, there was no effect of prey type, although barren grounds exhibited marginal differences with kelp forests (Wald $z = 2.032$, df residual = 343, $p = 0.0422$). Random effects were determined by fish species, although its variance was the smallest among the three previous analyses (Table S8). MaxN of *Scartichthys viridis/gigas* and *Pinguipes chilensis* followed a

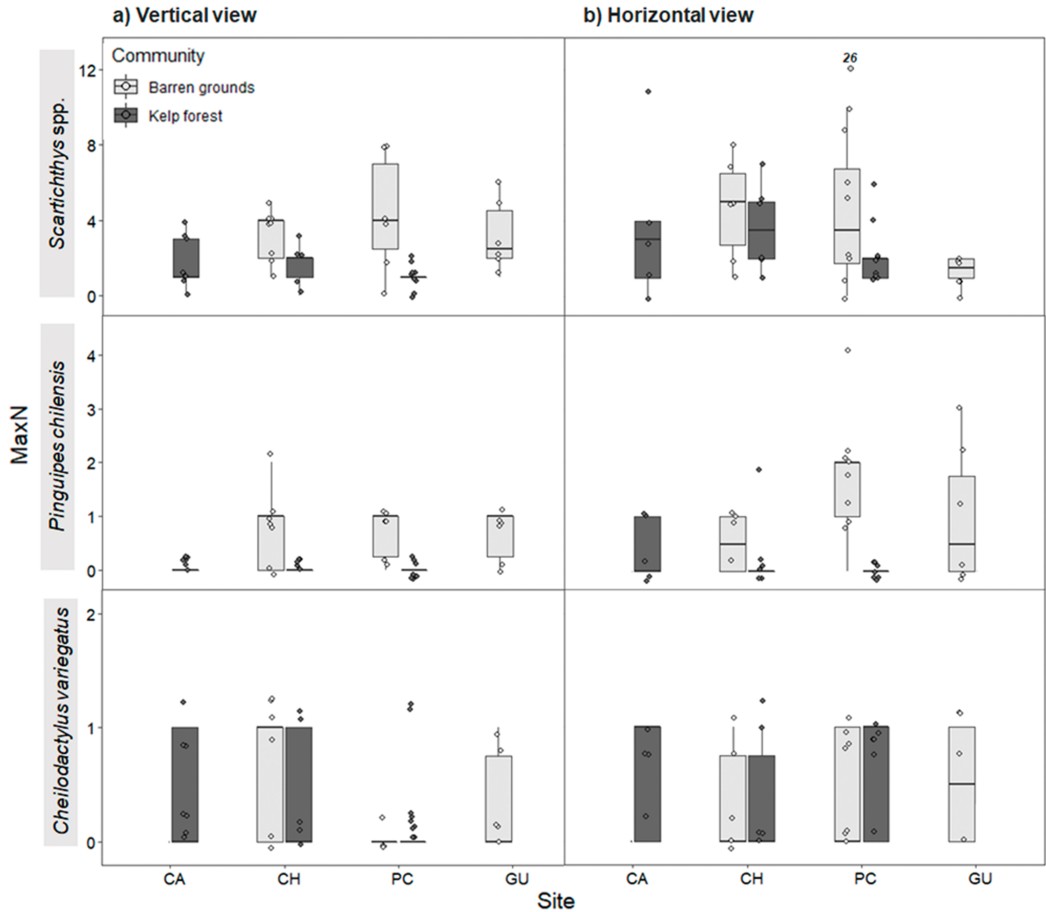

**Figure 6 Maximum abundance (MaxN) of the three most common predatory fish species by study site in videos with (A) vertical, and (B) horizontal view.** MaxN indicates the maximum abundance of any species observed in one of the 80 frames that were surveyed every 30 s during the first 40 min of the video recordings. The box plots present first and third quartile, while the bold lines in the boxes show the medians of MaxN; vertical lines are the minimum and maximum values (1.5*IQR or the interquartile range); dots show jittered raw data points and dots outside the range are outliers; dots with a number above them (which correspond to a MaxN value) represent outliers outside the y-axis scale.

pattern similar to FO (Fig. 6) and MaxN of *Cheilodactylus variegatus* was similar among study sites.

Even though the sizes of fishes were not consistently estimated in this study, all observed fishes were smaller than the plates (40 cm side length) to which prey items were tethered. The individuals of *Pinguipes chilensis* seen near the plates in videos were ~20–30 cm in body length. In the less abundant species *G. nigra*, all individuals were juveniles of ~30 cm. Also, during the deployment of structures, no large individuals of these species were seen by the divers.

## Predation assays and identity of predators

Predation pressure for both prey species did not vary across locations and habitats, but significant differences were observed between sea urchin and crab consumption. Predation of porcelanid crabs was significantly higher than that of sea urchins (Wald $z = -6.396$, df

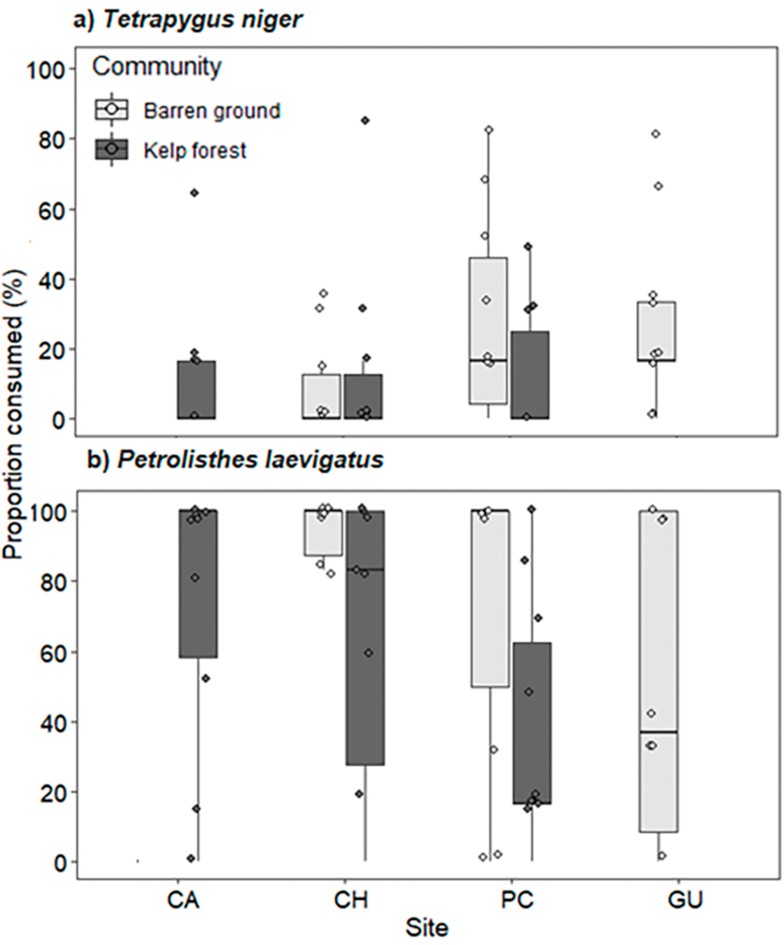

**Figure 7 Percentage of (A) *T. niger* (sea urchin), and (B) *P. laevigatus* (crab) consumed at each study site.** The box plots present first and third quartile, while the bold lines in the boxes show the medians of consumption (%); vertical lines are the minimum and maximum values (1.5*IQR or the interquartile range); dots show jittered raw data points and dots outside the range are outliers.

residual = 116, *p* < 0.001, Table S9). Consumption of *T. niger* ranged from 8.3% ± 14.2% on CH barren grounds to 29.6% ± 28.6% on GU barren grounds (Fig. 7A). In contrast, predation of porcelanid crabs ranged from 38.3% ± 34.3% in the PC kelp forest to 86.7% ± 31.2% on CH barren grounds (Fig. 7B).

*Scartichthys viridis/gigas* and *Cheilodactylus variegatus* were seen preying on bait in video recordings (Table S10). *Scartichthys viridis/gigas* was the principal consumer of crabs (82% of consumed prey) preying on 25 individuals of *Petrolisthes laevigatus* on barren grounds and seven individuals in kelp forests. *Scartichthys viridis/gigas* was also the only fish species recorded attacking (but not consuming) *T. niger*. It was observed in three instances turning over sea urchins (solitary individuals) and attacking them orally on CH and PC barren grounds. One instance each of predation by the brachyuran crabs *Homalaspis plana* and *Romaleon setosum* was observed, consuming sea urchins *T. niger* on PC barren grounds. Additionally, a predation event by the gastropod *Concholepas*

*concholepas* over *Petrolisthes laevigatus* was observed by the diver when cameras were removed (2 h after deployment), but this was not recorded in videos.

## DISCUSSION

Fish SR showed no major differences across locations and habitats, contrary to what was expected (higher richness in kelp forests). Values of FO and MaxN showed minor differences between habitats, which when observed, were higher on barren grounds than in kelp forests. These differences were mainly due to the presence of two abundant species *Scartichthys viridis/gigas* and *Pinguipes chilensis*, while other large fish predators occurred very infrequently. Predation on *Petrolisthes laevigatus* was significantly higher than on *T. niger*, possibly because crabs are a major diet component of coastal fishes, or because large predatory fish that could have consumed the sea urchins were mostly absent. *Scartichthys viridis/gigas* was the principal predator identified in this study, mainly consuming crabs; it was also the only fish that was observed attacking *T. niger*, confirming the predatory character of this species. Total consumption after 24 h was similar between habitats, coinciding with the minor or lack of differences in MaxN and FO. Our findings suggest that *T. niger* might have increased in abundances on subtidal hard bottoms, because current predation pressure on this species is low, possibly due to overall reduced abundances of large, top fish predators.

### Fish communities

Fish assemblages observed here were composed of typical species from the Chilean coast. The comparatively low total number of fish species per site was similar to SR of reef-fish observed by other studies along the central and northern coast of Chile (*Gelcich et al., 2008*; *Ory et al., 2012*; *Villegas et al., 2018*). *Pérez-Matus et al. (2007)* studied the structure of fish assemblages in northern Chile and found that CA was the site with the highest SR. This site was also the location with the highest total number of species in our study, but when comparing average SR per video among study sites, no differences were observed. SR did not vary across locations and habitats, except for vertical videos where SR was higher on barren grounds. This is not totally unexpected, because although kelp forests usually have higher diversity than other types of communities (*Levin & Hay, 1996*; *Graham, 2004*), studies in eastern Australia (*Curley, Kingsford & Gillanders, 2002*) and New Zealand (*Hesse, Stanley & Jeffs, 2016*) observed higher SR, abundance and predation pressure by fish on barren grounds than in kelp forests. On the other hand, *Angel & Ojeda (2001)* observed similar values in coastal fish abundances and diversity in northern Chile, when comparing a site with kelp forest and a site without forest.

*Scartichthys viridis/gigas* and *Pinguipes chilensis* were the most abundant species at all study sites, being more frequent on barren grounds albeit with small differences in horizontal videos (wider field of view). *Scartichthys viridis/gigas* are comparatively small fishes (10–20 cm adult size) that are not commercially fished (*Méndez-Abarca & Mundaca, 2016*) and are very common in subtidal habitats of northern Chile (*Villegas et al., 2018*; this study). On the other hand, *Pinguipes chilensis* was consistently present on barren grounds but not in kelp forests, which appears to be typical for this species, since

previous studies had observed that its abundances increase when density of kelp forests decreases (*Pérez-Matus et al., 2007*). The third most abundant predatory species was *Cheilodactylus variegatus*, which despite being more common in kelp forests (*Angel & Ojeda, 2001*; *Pérez-Matus et al., 2007*; *Villegas et al., 2018*) did not show statistical differences in this study.

Several fish species that are typical members of kelp forest communities in northern Chile were rarely seen in our study (e.g. *Hemilutjanus macrophthalmos*, *G. nigra*, and *Semicossyphus darwini*; *Pérez-Matus et al., 2012*), possibly due to low abundances caused by high fishing pressure (*Godoy et al., 2010*, *2016*). It should be noted that the method often used for the determination of SR are visual transect surveys (*Pérez-Matus et al., 2007*; *Ory et al., 2012*). Static recordings (video), as used in our study, could underestimate richness and abundances, because less abundant or less mobile species will be less frequently seen compared to visual transect surveys covering more extensive areas (*Pérez-Matus et al., 2007*). This could be partly responsible for the low abundances observed; indeed, we generally observed higher abundances in the horizontal videos with wider visual field. On the other hand, visual surveys may underestimate the presence and abundance of fish species with cryptic behaviours (*Willis, 2001*), such as *Scartichthys viridis/gigas*, the most frequent and abundant species in our study. Using both methods (visual transect surveys, and BRUVs) in the same study could provide the best estimates of SR and abundance of demersal fishes, since biology and behaviour of studied species could affect results when using different census methods (*Willis, Millar & Babcock, 2000*).

## Predation rates across communities

We observed no significant differences in predation rates between habitats for both prey species, which also coincides with findings of similar fish assemblages in kelp forests as well as on barren grounds. On the other hand, consumption was very different between the two prey species, with crabs being much more consumed than sea urchins. This was somewhat expected because crustaceans have a greater index of relative abundance in the stomach contents of temperate reef fishes (*Vargas, Soto & Guzmán, 1999*; *Medina, Araya & Vega, 2004*; *Pérez-Matus et al., 2012*), thus being preferred over sea urchins. However, although echinoderms are not the most consumed prey items, they are frequently found in the stomach contents of these fish. The more abundant predators (*Scartichthys viridis/gigas* and *Pinguipes chilensis*) were ineffective in consuming sea urchins, while the other less abundant species (e.g. *G. nigra*, *Semicossyphus darwini*, or *Cheilodactylus variegatus*) are recognised as important top predators along the Chilean coast (*Pérez-Matus et al., 2007*, *2017b*). Indeed, *Selden et al. (2017)* described a direct relationship between the rate of sea urchin consumption and the size of the predatory fish *Semicossyphus pulcher* in California. The lack of large individuals of *Semicossyphus darwini* and similar fish predators in our study area could thus be an explanation for low predation pressure on sea urchins, thereby ultimately being responsible for their proliferation and the establishment of extensive barren grounds along the coast of northern Chile.

Seasonality could also affect results since this study was done over a year. Seasonal temperature variations may affect metabolic rates of organisms (*Brown et al., 2004*; *Milazzo*

*et al., 2013*), and thereby influence their predation rates (*Duffy et al., 2015*). However, seasonal temperature variation along the exposed coast of Chile is rather moderate (*Thiel et al., 2007*) and might be insufficient to generate metabolic shifts in consumers as has been described for other latitudes (*Brown et al., 2004*). Furthermore, fish assemblages in the study region are similar across different seasons and fish species are characterised as being territorial and non-migratory (*Angel & Ojeda, 2001*; *Pérez-Matus et al., 2007*, *2012*). Consequently, it is unlikely that fish abundances and predation rates are affected by the season.

*Scartichthys viridis/gigas* was the species that consumed the most prey during video recordings. This species has originally been described as an herbivorous species with more than 90% of their stomach contents comprising seaweeds, mostly chlorophytes and rhodophytes (*Ojeda & Muñoz, 1999*; *Muñoz & Ojeda, 2000*). However, those studies were conducted on individuals from the intertidal zone, with little information on the foraging behaviour of this species in subtidal habitats. *Ory et al. (2012)* observed effective predation events of *Scartichthys viridis* on *Rhynchocinetes typus* in the shallow subtidal zone: the initial attack of an individual, usually adult, triggers a series of joint attacks by multiple conspecifics, forming a kind of feeding frenzy, which ends when the prey item is completely consumed. Similar observations had already been reported by *Dumont, Gaymer & Thiel (2011)*, classifying *Scartichthys viridis* as the most efficient predator on the ascidian *Ciona intestinalis* within the Coastal System of Coquimbo (30°S). Our study not only confirms active predation but also group foraging in *Scartichthys viridis/gigas*. However, similar to reports by *Urriago, Himmelman & Gaymer (2012)*, *Scartichthys viridis/gigas* attacked and damaged *T. niger* but did not achieve successful predation in our videos. Whether the frenzy generated by this blennid fish might attract other predators of sea urchins is not known at present.

*Cheilodactylus variegatus* was the only other predatory fish preying on our baits, specifically on crabs, which is consistent with previous studies that observed porcelanid crabs in the diet of this species (*Angel & Ojeda, 2001*; *Palma & Ojeda, 2002*; *Pérez-Matus et al., 2012*). There are several other fish species that could have preyed on our baits, such as *G. nigra*, *Semicossyphus darwini*, *Hemilutjanus macrophthalmos*, and *Pinguipes chilensis*, among others (*Muñoz & Ojeda, 1997*; *Medina, Araya & Vega, 2004*; *Pérez-Matus et al., 2012*). These species, which could potentially be predators and controllers of sea urchins, were not identified as predators, most likely because they were only observed in low abundances in our study (except for *Pinguipes chilensis*). Possibly, low abundances and the small sizes of adult fishes observed in our videos (see also *Selden et al., 2017*) are the main reason for not recording them to prey on our baits. On the other hand, some of the species that consumed our prey (during the 24 h of the assay) may have nocturnal behaviours and were thus not captured by our diurnal video recordings.

In addition to fish predators, seastars *Heliaster helianthus* and *Meyenaster gelatinosus* are among the main consumers of sea urchins (*Gaymer & Himmelman, 2008*; *Urriago, Himmelman & Gaymer, 2011*, *2012*), but in our study they seemed to be unimportant, not being interested in the tethered prey. On the other hand, brachyuran crabs *Homalaspis plana* and *Romaleon setosum* were observed attacking (and successfully preying on)

tethered sea urchins. While our study is focused on fish predators, crabs also might be important controllers of black sea urchins, an interpretation that is supported by previous studies that have recognised these species as consumers of echinoids (*Morales & Antezana, 1983*; *Cerda & Wolff, 1993*). Future studies of subtidal predator assemblages should also take carnivorous crabs into account, because they seem to have an important role in subtidal rocky reefs.

## CONCLUSION

We observed no differences in predation pressure between habitats, which could be related to the similar SR and abundances of the observed fishes. We also recorded very low abundances of important top fish predators at all sites (e.g. *Semicossyphus darwini*), which might be due to overfishing (*Godoy et al., 2010*, *2016*) or due to the method used to count (stationary videos). The overall lack of large fish predators may also facilitate small and opportunistic predators such as *Scartichthys viridis/gigas* (not subject to fishing), which have been observed in this study as important predators in both habitats, kelp forests and barren grounds. The fact that these small fish were frequently observed in open areas during bright daylight furthermore suggests that large predatory fish are rare in the study area, which could also have favoured sea urchin survival and spreading on subtidal hard bottoms. In summary, high fishing pressure and the consequent lack of top fish predators may have profound effects on shallow subtidal communities along the Chilean coast. This needs further attention, considering the ecological importance of kelp forests and their increasing exploitation (*Almanza & Buschmann, 2013*; *Buschmann et al., 2014*).

## ACKNOWLEDGEMENTS

We are foremost grateful to the fishermen who allowed us to work in their AMERBs. We thank the boat people, and especially Freddy Gonzalez, for their help during this study. We are very grateful to Miles Abadilla for editing the final manuscript. We also thank Boris López, who helped with the maps in fig. 1.

### Funding

Funding was received through a FONDECYT grant (CONICYT-FONDECYT 1161383). The funders had no role in study design, data collection and analysis, decision to publish, or preparation of the manuscript.

### Grant Disclosures

The following grant information was disclosed by the authors:
FONDECYT grant: CONICYT-FONDECYT 1161383.

### Competing Interests

The authors declare that they have no competing interests.
## Author Contributions

- Nicolás Riquelme-Pérez conceived and designed the experiments, performed the experiments, analysed the data, prepared figures and/or tables, authored or reviewed drafts of the paper, approved the final draft.
- Catalina A. Musrri analysed the data, prepared figures and/or tables, authored or reviewed drafts of the paper, approved the final draft.
- Wolfgang B. Stotz conceived and designed the experiments, contributed reagents/materials/analysis tools, authored or reviewed drafts of the paper, approved the final draft.
- Osvaldo Cerda analysed the data, authored or reviewed drafts of the paper, approved the final draft.
- Oscar Pino-Olivares conceived and designed the experiments, performed the experiments, contributed reagents/materials/analysis tools, approved the final draft.
- Martin Thiel conceived and designed the experiments, performed the experiments, contributed reagents/materials/analysis tools, authored or reviewed drafts of the paper, approved the final draft.

## Field Study Permissions

The following information was supplied relating to field study approvals (i.e. approving body and any reference numbers):

Surveys were conducted under the resolution 2,649 from the 'Subsecretaría de Pesca y Acuicultura' (SUBPESCA) published in the 'Official Newspaper of Chilean Republic', num. 41,553 on 7 September 2016.

## Data Availability

The raw data is available in the Supplemental Files.

## Supplemental Information

Supplemental information for this article can be found online at http://dx.doi.org/10.7717/peerj.6964#supplemental-information.

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
