# Peer review of "Coastal fish assemblages and predation pressure in northern-central Chilean Lessonia trabeculata kelp forests and barren grounds"

_PeerJ, doi:10.7717/peerj.6964_

## Round 0.1 · original submission · Major Revisions

As you will see below, the reviewers have provided an extensive list of very constructive comments to improve the quality of the final Manuscript.

In particular, reviewers 1 and 2 urge you to reconsider some of the main inferences in the Discussion, and urge you to restructure your text to exercise more caution in your conclusions. Reviewer 1 also raises some important questions regarding seasonality and independence among camera replicates and Reviewer 2 also suggests a re-analysis of the data on the basis of different kelp density.

·

Basic reporting

While the English is generally very good there are areas that could be improved for clarity (see below). It is also worth noting that you can generally get away with not putting ‘the’ in front of a large number of words. I generally read aloud and if it doesn’t affect meaning get rid of ‘the’. The authors also use ‘herein’ throughout the manuscript and this at times isn’t helpful or breaks up flow. I would suggest not using this term.

The introduction provides a good background to the study. Specific comments below:
Line 50 heatwaves is one word
Line 53 Add the before absence
Line 57 Suggest changing to Grazing activity is…..
Line 58 Delete These
Line 66 Suggest changing sentence structure to “ In these cases, high predator abundances control grazers leading to increased algal biomass.
Line 70 Delete ‘however’ and replace with “Outside of protected areas….
Line 76 Delete “mentioned as” and replaced with “considered”
Line 78-80 It is not clear to me why there is discussion of sea stars when they are not the focus of the study. Moreover, given their role of predators of urchins it is interesting that they are not mentioned being potentially responsible for predation observed in this study. Were these species observed by the videos or are they present at the study locations?
Line 83-86 Suggest restructuring this sentence to “All these species have been subject to intense fishing activity with population declines…..
Line 88 Are the authors suggesting that the role of fish predation in controlling urchin numbers is less well known? I suggest that the authors clarify this point.
Line 89-91 This sentence needs a reference

The figures and tables are relevant, but I feel that more information is required in many of the legends and the figures 4-7 are messy in places.
Table S1 KF and BG need to be defined in the table legend. Also I am guessing that O should be W for a paper written in English
Figure 1 The authors might consider including different coloured symbols to denote different management regimes
Figure 4-7 The authors need to explain what the point symbols represent (I am guessing raw data). I also found their placement a little random. Have they been jittered? If so I wonder if it would be better to jitter in the vertical rather than the horizontal? Also the lines, boxes and error bars need to be defined to enable the reader to interpret the data.
Fig 5 & 6 I suggest the authors explain that the numbers on the outliers represent FO and Nmax respectfully.
Fig 7 Suggest rephrasing to Percentage of a) T. niger (urchin) and b) P. laevigatus (crab) consumed at …….

The structure conforms to peerj standards and I can confirm that the raw material has been supplied

Experimental design

The research question is well defined and the methods are generally described in enough detail (but see specific comments on how to improve clarity). The design is adequate, although not ideal and the authors need to be careful how they interpret the results. For example, there are sites where both kelp and barrens areas are located within 1 km and then an additional two sites: one a barrens site and one a kelp site. Obviously a more balanced design with site pairs would have been better.

The study seeks to compare fish richness and the FO and NMax of the three most abundant species. They do this by using baited video (analysing 80 still frames over a 40 minute period) that are positioned horizontally and downward. In addition they tether two prey items (urchins and crabs) to plates within the field of view of the videos. Video footage is used to identify species removing the prey items (only 40 mins of film) and then predation is measured after 24 hours.

The data for the study is collected over two days at each sites, but the full dataset is collected over one year with data collected across all seasons. The authors mention seasonality in the discussion and given that fish vary so much in space and time I suggest that the authors provide some evidence base in the methods/discussion to suggest that while the data were collected over very different seasons that the data is still comparable because the fish they were studying are non-migratory or don’t vary in terms of abundance through time. It might be worth the authors stating that any such migratory fish were excluded.

My other major concern regards the placement of the cameras. The authors state that this is within 10m of each other, which is very close and certainly means that fish would have been able to swim between cameras therefore invalidating assumptions of independence. I believe this deployment needs some justification.

Specific comments
Line 99 Suggest starting sentence “The study…” as this isn’t describing the predation assays
Line 116 What does deep-type refer to in the context of barrens?
Line 118-120 Two study sites were separated by 500m - to 2km. It would be useful for the authors to explain and provide an evidence base for whether fish identified in the study could move between these sites.
Line 120 Remove miles around and change to kilometres
Line 145 Delete to take and replace with allowed images to be taken….
Line 147 Delete allowed taking and replace with took
Lines 149-153 It would be useful to include references to support the statement made regarding the camera set up and pros and cons. I also think the authors need to make it clear that the horizontal camera only provides information within the field of view rather than providing information for the whole surrounding area.
Line 154 Should be mesh. And what is an eternity plate?
Line 158 Add us after allowed
Lines 160-162 Suggest the authors restructure. At each study site, five vertical and five horizontal structures were randomly distributed.
Line 167 Suggest here and throughout that the authors refer to the crabs and urchins as prey items rather than bait.
Lines 167-170 The authors need to make it clear that this design was not used at BG.
Line 180 How did the authors determine where 2m was in the field or view?
Line 183 Often people exclude a period of time at the beginning of the video because of potential disturbance in placing the cameras. Are the authors sure that disturbance may not have affected the early frames used in their study.
Predation assays - I think it is important that the authors make it clear that there is only video evidence of predation events for the first 40 mins. After this period the authors do not have a definitive knowledge of which species was responsible for predation events after this time.
Line 204 Change to monofilament fishing line
Data analysis – The authors need to make it clear to the reader that the two days data were combined for all analyses. Also I suggest that the authors refer to habitat (2 levels: kelp forest and barrens) instead of community as it is a little confusing.
Line 228 I am a little confused with the design used for the FO and Nmax analysis. Where does species list and replicates come from?

Validity of the findings

I believe the authors need to be very careful with the interpretation of their findings. There are a number of statements made in the discussion, which are just not backed up by the data. I therefore think that the discussion needs a significant restructure to reflect what can be said based on the outcomes of this research. The results, as presented, suggest only limited significant effect i.e. one site being higher lower than another rather than a consistent response across habitats. This therefore means there is little support for some of the general statements which suggest consistent patterns. I believe the authors need to honest regarding their data and admit the patterns are not hugely clear.

Lines 343-362 Discuss how no take zones can lead to increased richness. However, the data presented in this paper doesn’t support this. The one protected zone did not have richness, FO or Nmax compared to the fished sites so the statements made here are not justified by the data.
Lines 363-369 There was no difference in predation on barren versus kelp areas therefore it is impossible to say that prey were more accessible on barren grounds
Lines 371-375 I really don’t think the data presented in Fig 6 supports the statement for a slight tendency for higher FO in kelp forests. There was no significant difference and FO in kelp forests were just highly variable.
Lines 390-391 Again this pattern is not general there was only one significant pair wise comparison between habitats for the two most common species with no differences across other sites and habitats. For the third most abundant there was no effect of either location or habitat.
Lines 407-410 This needs to toned down as it wasn’t tested for in this study. Is information known on what size fish take the urchins put out in this experiment?
Lines 411-418 The authors raise issues regarding seasonality and metabolic rates, but more importantly highlight that because data was collected at different times of the year it could have impacted on the observed responses. The issue of sampling during different time periods needs addressing and especially whether it could impact expectation regarding measures of richness, FO and Nmax.
For all stats in the results only the p-values are provided. This isn’t enough information and I suggest that the authors either include in-text stats information or provide output tables in the SOM that are referred to in the main text. Also it appears that there were only limited significant post-hoc pair-wise comparisons. In places it is cleary stated that there were no other significant differences. I just encourage the authors to ensure this is the case everywhere.
Abstract Line 35-38 I suggest the authors need to rephrase this last sentence as it is just not supported by the data presented.

Minor comments
Lines 253-261 I suggest that the authors make it clear that some of the fish mentioned were only found at one site. I think the last sentence is a little strong when the authors are presenting qualitative data and the difference is only once species. When this data is formally analysed there is no significant difference.
Line 262-267 I was surprised at the low richness in all sites and habitats. It came as a surprise to me and I wonder if it is worth just making this general statement.
Line 271 Refer to the figure
Line 275-278 Surely the main test is the most powerful and if your post-hoc tests of a significant main test are coming back non-significant than the post-hoc tests you are using is too conservative.
Line 298 Suggest restructuring sentence to “Predation pressure for both types of prey items did not vary across locations or habitats
Lines 304-306 Suggest restructuring this sentence for greater clarity

Additional comments

The research presented is interesting, but I just think that the results have been over interpreted.

Reviewer 2 ·

Basic reporting

The present paper aims to understand he differences in predation pressure in two contrasting habitat types commonly occurred in temperate reefs (i.e., kelp forests and urchin-dominated barren grounds). Authors compare species richness, abundance and predation impacts of predatory reef fishes in the two contrasting habitat types using underwater video analysis. While the subject of interests matters in terms of potential mechanism that may underlie community shifts in kelp forest ecosystems, I consider that the manuscript have potential to be published after improving some areas and aspect of the study which I will summarize below: a) the problem (motivation) raised in the introduction, need better arguments centered in the hypothesis tested; b) this is linked to the materials and methods (hence experimental design and overly analyzed data) where I found that the manuscript has some editorial problems in terms of simplifying what wants to explain particularly. there are many paragraphs explaining something that could easily be explained in one short paragraphs; c) I found where most work has to be done in order to provide a compelling evidence of a lack of difference in these contrasting habitat types is in explaining and justifying the methodology and data analyzes: 1) vertical views versus horizontal views of data frame, insufficient number of cameras, 2) MxN of 30 second per frame do not avoid double counting of reef fishes (please see Malcolm, H., & Speare, P. (2003). Potential of video techniques to monitor diversity, abundance and size of fish in studies of marine protected areas. Aquatic Protected Areas-what works best and how do we know, 455-464), and 3) selected tethered prey, one prey requires large carnivores to enter your small experimental arena and there are no potential cage (or structures built in controls) and as much as I acknowledge (to be honest) to have plenty of my own articles cited in this manuscript (except one, Perez-Matus et al. 2016 MEPS where we compare, among other stuff, tethering of cryptic reef fish for a short period of time in order to avoid escapes and finding differences in predation pressure in kelp versus non kelp habitat) I found invaluable a critique of tethering experiments to estimate predation pressure in the field without sensitivity (behavioral analysis) (please refer to Aronson RB, Heck KL Jr (1995) Tethering experiments and hypothesis testing in ecology. Mar Ecol Prog Ser 121: 307−309). These points raised above definitely leads to d) interpretation of data, although I found the discussion a bit lengthy authors tried to explain their results.

As a final comment I feel that they have a good working material (e.g handling time, effective consumption, first arrival time to the experimental arena) but authors are not taking all the relevant information that can be obtained from them and they should take advantage of footage and comments such as in lines 256-258 and other should be incorporate in the fancy statistical procedure (will get back to that shortly).

Experimental design

The experimental design is straight forward, I know how hard it is to work in the system and having few o more replicated means a lot of work in the field, but how many cameras (replicates did you used? at what depth?) But given the great difference between the sites in terms of density and morphologies of the kelp forest, I think that the work should be re-analyzed not in two categories only (KF and BG) but in factors of density of the kelp (probably three categories of this). There are some flaws in explaining how data was analyzed and the rationale of using such as framework (using replicates as random parameter to avoid over dispersion is somewhat misleading) and lastly I do not understand the justification of using two techniques to estimate abundance. I think that each one allows to obtain different useful information for the work. In terms of The generalized linear mixed effects model (GLMM) approach need to be better described as it seems that you conducted two different analyses (see lines 217-241). I still think that I feel that it is not appropriate for the data. I would like to see a very concise description of the model validation (see Alan Zurr and colleagues outline these nicely). I would expect to see the following checks: Check for outliers and influential points, linearity between the predictor variables and the quasipoisson- or quasibinomial-transformed response variable, Check for multicollinearity among predictor variables by, for example, variance inflation factor (VIF) analysis, Check for spatial autocorrelation of model residuals using, for example, spline correlograms. In reference to the GLMMs, what is meant by “sample replicates treated as random effects to avoid autocorrelation", if so these should be nested in locations? I think that site need to be treated as a random effect due to site-level clustering of the data ( I will suggest doing that) and the degree of clustering should be reported by, for example, the intraclass correlation coefficient (ICC) or the Variance Partition Coefficient (VPC) variance of higher level divided by the sum of variances of site and the higher-level variance of our predictor variable.
If using GLLMs, need to report the residual df on the model df, also the effect sizes and p - values. I think you don't have much zeros so poisson error is ok, but pressure over dispersion. I recommend you use all the length of the videos, specify the rationale of using vertical and horizontal views (there are in different scales), speared frames for 5 min (or more than a min) to avoid double counts. Also if you know the scale of you trapezoid frames, in your plates you can calculate the size of the fish (see lines 292-293) and then convert to biomass (see Perez-Matus et al. 2014 RBMO).

Validity of the findings

Much of the introduction in centered on the community effect of predation of urchins but few on the use of the porcelain crab which is not a grazer. I think this need to be justified as per lines 206-208 This would serve for a prey preference analysis, however both have very different exposures to their predators in the environment. For the rest, since it is not part of the question, the most convenient and informative would have been using grazer species that depredate the kelp, and therefore a niche overlap with the other tethered prey. Let, in terms of tether how can you estimate the escapes?

Need to be explained why horizontal versus vertical views
Avoid some redundancy in methods (see lines 163)
Also, richness was evaluated but no rationale explained for that, in lines 176 was conveyed that not all species were counted but some reported.

Additional comments

I have tried to be constructive in my assessment of the ms, I would like to see this study out but a re-analysis of the videos and the data can be helpful to provide information about te predation pressure and differences between BG vs KF. I consider your paper valuable as most of the fishing on KF may potentially be transformed to BG s and data is valuable in terms of the interactions and potential prediction of maintenance of contrasting communities.

Good luck in revising this manuscripts

Reviewer 3 ·

Basic reporting

The authors present a nice experimental approach to study the effect of fish on grazers to explain contrasting communities (barren grounds vs kelp forests). In general, the authors made a good job managing the existing literature, but some local literature are needed at least from my perspective in different subjects:
i) The authors must cite a highly relevant kelp-grazing paper for Chile Dayton 1985 Ecol Monogr Vol 55
ii) The discussion should mention that Tegula atra can reduce kelp recruits but also other competing algae (Henríquez et al. JPhycol 2011)
iii) There are previous studies comparing recruitment success between non-kelp kelp habitats (Almanza et al. Mar Biol Res 2012 and Almanza et al. 2013 Int J Env and Sust Dev)
In addition, the authors need to be more precise in some statements as the authors indicate that:
i) The authors indicate “Kelp forest are declining globally” citing Krumhansl et al 2016 (Line 47-48) which is not completely the case as only a proportion of the places in Chile is showing this trend and actually in Chile only in the central part of the country.
ii) The authors indicate that “In the absence of forests there are extensive patches of ….. called barren grounds”. The paper cited ibn the same study (Filbee-Dexter & Scheibling 2014) show that and alternative state is produced monopolized by turf algae which are not a barren ground.
In addition, please add in the figure legends what are the box plots and dots indicating (SD, range, other?).
Statistical analysis. In the text the authors provide only then the p values (e.g. lines 282 and 283). There is a need to provide more detailed statistical information. Add a table or include it in the parenthesis.
Finally, the Abstract must be modified. The first sentence is not completely valid (see Filbee-Dexter & Scheibling 2014) and the second sentence must also take into account that human action can be part of the explanation and not only the direct impact of grazers (this is acknowledged in the introduction (line 52) but the authors should not bias the abstract).
Some additional minor elements:
i) Line 73; provide the Lessonia species name,
ii) Line 176; The authors use abbreviation FO and MaxN but not for species richness. If the authors use abbreviations for these variables used for the 3 ones or for none of them.

Experimental design

The experimental design and the number of study sites is acceptable considering the logistic difficulties to carry out this type of studies. I only demand some more detailed statistical information in the text.

Validity of the findings

The finding's that this manuscript provide about the relevance of barren grounds are valid and offer an explanations in relation to the importance of barter grounds for foraging. The Conclusion section is clear for me.

Additional comments

By answering the points raised in the "Basic reporting" section the paper should be ready for a final review.

---

## Round 0.2 · accepted · Accept

I have reviewed your response to the thorough and constructive reviews and believe you have addressed all the points raised and the MS has consequently been improved as a result.

Please see a couple of typos that I picked up when re-reading your reviewed MS in the attached pdf, plus a comment about fish species diversity/ abundance also being higher in SE Australia.

Congratulations!

#